# Stable G protein-effector complexes in striatal neurons: mechanism of assembly and role in neurotransmitter signaling

Keqiang Xie[1], Ikuo Masuho[1], Chien-Cheng Shih[1,2], Yan Cao[1], Keita Sasaki[3], Chun Wan J Lai[4], Pyung-Lim Han[5], Hiroshi Ueda[3], Carmen W Dessauer[6], Michelle E Ehrlich[7,8,9], Baoji Xu[1], Barry M Willardson[4], Kirill A Martemyanov[1]*

[1]Department of Neuroscience, The Scripps Research Institute, Jupiter, United States; [2]Department of Pharmacology and Physiology, Georgetown University Medical Center, Washington, United States; [3]Department of Pharmacology and Therapeutic Innovation, Graduate School of Biomedical Sciences, Nagasaki University, Nagasaki, Japan; [4]Department of Chemistry and Biochemistry, Brigham Young University, Provo, United States; [5]Department of Brain and Cognitive Sciences, Ewha Womans University, Seoul, Republic of Korea; [6]Department of Integrative Biology and Pharmacology, The University of Texas Health Science Center, Houston, United States; [7]Department of Neurology, Icahn School of Medicine at Mount Sinai, New York, United States; [8]Department of Pediatrics, Icahn School of Medicine at Mount Sinai, New York, United States; [9]Department of Genetics and Genomic Sciences, Icahn School of Medicine at Mount Sinai, New York, United States

*For correspondence: kirill@scripps.edu

Competing interests: The authors declare that no competing interests exist.

**Abstract** In the striatum, signaling via G protein-coupled neurotransmitter receptors is essential for motor control. Critical to this process is the effector enzyme adenylyl cyclase type 5 (AC5) that produces second messenger cAMP upon receptor-mediated activation by G protein Golf. However, the molecular organization of the $G_{olf}$-AC5 signaling axis is not well understood. In this study, we report that in the striatum AC5 exists in a stable pre-coupled complex with subunits of $G_{olf}$ heterotrimer. We use genetic mouse models with disruption in individual components of the complex to reveal hierarchical order of interactions required for AC5-$G_{olf}$ stability. We further identify that the assembly of AC5-$G_{olf}$ complex is mediated by PhLP1 chaperone that plays central role in neurotransmitter receptor coupling to cAMP production motor learning. These findings provide evidence for the existence of stable G protein-effector signaling complexes and identify a new component essential for their assembly.

## Introduction

Neurotransmitters elicit their effects by activating receptors on the surface of neurons. G protein-coupled receptors (GPCRs) form the largest group of the receptors responsible for the actions of the majority of neurotransmitters and play a critical role in virtually all neuronal functions (*Gainetdinov et al., 2004*; *Wettschureck and Offermanns, 2005*). In a classical model, upon binding to neurotransmitter, GPCRs undergo conformational changes activating heterotrimeric G proteins by promoting GTP binding to Gα subunits and triggering the release of the Gβγ subunits. When dissociated, both Gα and Gβγ subunits modulate the activities of downstream effector molecules that are directly responsible for generating cellular responses (*Gilman, 1987*; *Neer, 1995*).

**eLife digest** In the brain, cells called neurons communicate with one another using chemicals called neurotransmitters, which bind to receptors on the surface of cells. In most cases, the binding of neurotransmitter causes a protein known as a G protein to interact with the receptor. G proteins are made up of three subunits: α, β and γ. After binding to the receptor, the α subunit separates from the β/γ subunits – which remain together – and both components then act as signals to activate specific targets in the cell.

There are many different α, β and γ subunits, which participate in various signaling pathways in different parts of the brain. In the striatum – a region involved in controlling movement – a particular combination of α, β and γ subunits called $G_{olf}$ is responsible for activating an enzyme called adenylyl cyclase type 5 (AC5).

Traditionally, G protein signalling has been thought to occur in stages so that the binding of neurotransmitter to receptors on striatal neurons would lead to the dissociation of $G_{olf}$ into the α and βγ subunits. Then the α and βγ subunits are thought to bind to and activate AC5. However, Xie et al. now show that all three subunits of $G_{olf}$ are already found in a stable group (or complex) with AC5 in striatal neurons. Neurotransmitter binding to the receptors causes the entire $G_{olf}$-AC5complex to rearrange and this process activates AC5.

A particular chaperone protein regulates the assembly of the G protein-AC5complex. Mice that lack the gene that encodes this chaperone in striatal neurons struggle to learn how to balance on a rotating rod. In humans, mutations in the genes that encode $G_{olf}$ and AC5 cause dystonia, which is a disorder characterised by involuntary movements. Given the evidence linking this chaperone protein to the regulation of $G_{olf}$-AC5 signaling, future experiments should investigate whether it might also contribute to dystonia.

However, an emerging alternative model suggests that G protein heterotrimers exist in more stable complexes that rearrange rather than dissociate upon activation and may further form higher order signaling complexes with receptors and effectors (*Bunemann et al., 2003*; *Dupre et al., 2009*; *Hepler, 2014*; *Lambert, 2008*). The assembly of this macromolecular complex may be tightly regulated. Indeed, several chaperone proteins have been described to be required for the biogenesis of the G protein subunits and their complexes (*Dupre et al., 2009*; *Papasergi et al., 2015*; *Willardson and Tracy, 2012*).

One of the central and best studied G protein effectors is adenylyl cyclase (AC), an enzyme that catalyzes the synthesis of the second messenger cyclic adenosine monophosphate (cAMP) (*Taussig and Gilman, 1995*). Numerous isoforms of AC are differentially modulated by both Gβγ and various Gα-GTP subunits and play critical roles in a variety of fundamental neuronal processes (*Sadana and Dessauer, 2009*; *Sunahara et al., 1996*). GPCR signaling to AC performs a particularly important function in the striatum, the input structure of the basal ganglia circuit essential for initiating and maintaining movement, mood control, and reward valuation (*Graybiel, 2000*; *Kreitzer and Malenka, 2008*). Imbalance in cAMP homeostasis in this region has been associated with drug addiction, bipolar disorder, schizophrenia and a variety of movement disorders (*Bonito-Oliva et al., 2011*; *Girault, 2012*; *Nestler and Aghajanian, 1997*; *Wilson and Brandon, 2015*). Striatal neurons receive diverse inputs that converge on AC5, the major AC isoform in the region, accounting for ~ 80% of cAMP generation (*Lee et al., 2002*). Coupling of key neurotransmitter receptors, such as dopamine D1 (D1R) and adenosine A2A (A2AR) to increase cAMP production in these neurons, is mediated to a large extent by a unique heterotrimer composed of the stimulatory α subunit $G\alpha_{olf}$ complexed with $G\beta_2$ and $G\gamma_7$ subunits (*Herve, 2011*). Indeed, deletion of $G\alpha_{olf}$, $G\gamma_7$ or AC5 in mice severely diminishes cAMP production in response to D1R and A2AR activation, which is paralleled by muted behavioral responses to psychostimulants and antipsychotics that act on these receptors as well as by profound motor deficits (*Corvol et al., 2007*; *Iwamoto et al., 2003*; *Lee et al., 2002*; *Sasaki et al., 2013*; *Schwindinger et al., 2010*). Recently, mutations in $G\alpha_{olf}$ and AC5 have been shown to cause primary dystonia in humans (*Carapito et al., 2014*; *Fuchs et al., 2012*; *Kumar et al., 2014*), further supporting the key contribution of $G_{olf}$-AC5 signaling axis to pathophysiology of movement disorders. These observations argue for the lack of functional compensation from other

AC isoforms and heterotrimeric G proteins in transducing the signal. However, the functional significance of striatal-specific composition of these specific signaling elements is not well understood.

Functionally, $G\alpha_{olf}$ is similar to another stimulatory G protein $G\alpha_s$, but $G\alpha_{olf}$ has lower efficiency of both receptor coupling and AC5 stimulation when tested in in vitro biochemical assays (*Chan et al., 2011*; *Jones et al., 1990*). AC5 displays dual regulation by G protein subunits in which $G\beta\gamma$ acts to facilitate its activation by $G\alpha_s$ (*Gao et al., 2007*). In vitro biochemical studies show that AC5 is capable of binding $G\alpha_s$ and $G\beta\gamma$ simultaneously, suggesting that it can scaffold the stimulatory G protein heterotrimers (*Sadana et al., 2009*). In fact, growing evidence suggests that in vivo $G_{olf}$-AC5 may exist in a signaling complex (*Herve, 2011*). In mice, AC5 elimination leads to a reduction in expression levels of $G\alpha_{olf}$ in the striatum (*Iwamoto et al., 2004*). Similarly, knockout of $G\gamma7$ reduces the expression of $G\alpha_{olf}$ and $G\beta_2$ (*Sasaki et al., 2013*; *Schwindinger et al., 2010*). However, interactions involving elements of the complex in the striatum, their reciprocal relationship, mechanisms of complex assembly as well as implications for the cAMP signaling and behavior are not understood.

Here we report that in the striatum, AC5 forms a stable macromolecular complex with heterotrimeric Golf proteins and this pre-assembly is essential for the stability of AC5 and its ability to produce cAMP. We identified that the $G\beta$ chaperone phosducin-like protein 1 (PhLP1) plays a key role in the assembly of this signaling complex in striatal neurons. Elimination of PhLP1 in striatal neurons affects assembly and stability of the complex and causes selective impairment in sensorimotor behavior and motor skill learning, preferentially affecting signaling in the striatopallidal medium spiny neurons.

## Results

### Stability of AC5 in the stratum depends on its complex formation with the Golf heterotrimer

To begin testing the idea that AC5 may exist in a stable complex with the Golf heterotrimer in vivo, we first analyzed their binding by co-immunoprecipitaiton assays. AC5 was effectively and specifically pulled down together with both $G\alpha olf$ and, to a lesser extent, $G\beta_2$ from mouse striatal lysates, indicating that in striatal neurons Golf subunits form stable complexes with AC5 in their ground state in the absence of receptor stimulation (*Figure 1A*). Next, we analyzed the relationship between AC5 and Golf by examining the co-dependence of their expression using mouse knockout models. Elimination of AC5 in mice (*Adcy5$^{-/-}$*) dramatically reduced the expression of $G\alpha olf$ but had no appreciable effect on $G\beta2$ expression, suggesting that AC5 contributes to the stability of $G\alpha olf$ but not the stability of the $G\beta_2\gamma_7$ complex (*Figure 1B*). We further used *Adcy5$^{-/-}$* striatal tissues in the immunoprecipitation experiments and found that antibodies against AC5 failed to pull-down $G_{olf}$ subunits from striatal lysates lacking AC5, confirming the specificity of $G_{olf}$ association with AC5 (*Figure 1C*). In contrast, disruption of $G\beta_2\gamma_7$ by eliminating $G\gamma_7$ in *Gng7$^{-/-}$* knockout led to a marked down-regulation of both AC5 and $G\alpha olf$ expression (*Figure 1D*). The consequences of $G\alpha_{olf}$ disruption were evaluated in heterozygous mice (*Gnal$^{+/-}$*) with severely diminished $G\alpha_{olf}$ expression because complete ablation of $G\alpha olf$ leads to increased perinatal lethality (*Belluscio et al., 1998*). In these animals, the levels of both AC5 and $G\beta_2$ were substantially reduced (*Figure 1E*). Therefore, in mouse striatum co-dependence of the complex components appears to be unidirectional: destabilization of either the $G\alpha$ or $G\beta\gamma$ subunits of the Golf heterotrimer compromised AC5 stability, while only $G\alpha$ but not $G\beta\gamma$ was affected by the loss of AC5.

We further examined the relationship between Golf subunits and AC5 in transfected HEK293 cells. Consistent with the in vivo data, co-expression with $G\alpha_{olf}$ dramatically increased levels of AC5, and vice versa, the introduction of AC5 enhanced the expression of $G\alpha_{olf}$ (*Figure 1F*). In contrast, no significant changes in AC5 or $G\beta_2\gamma_7$ levels were observed upon their co-expression (*Figure 1G*). We further observed co-dependence between $G\alpha_{olf}$ and $G\beta_2\gamma_7$ subunits in which co-expression with $G\alpha_{olf}$ increased the levels of $G\beta_2\gamma_7$ and co-expression of $G\beta_2\gamma_7$ enhanced levels $G\alpha_{olf}$ (*Figure 1H*). Collectively, these data strongly support the existence of a complex between AC5 and the $G\alpha_{olf}\beta_2\gamma_7$ heterotrimer and indicate that the association between components of this complex is required for its stability.

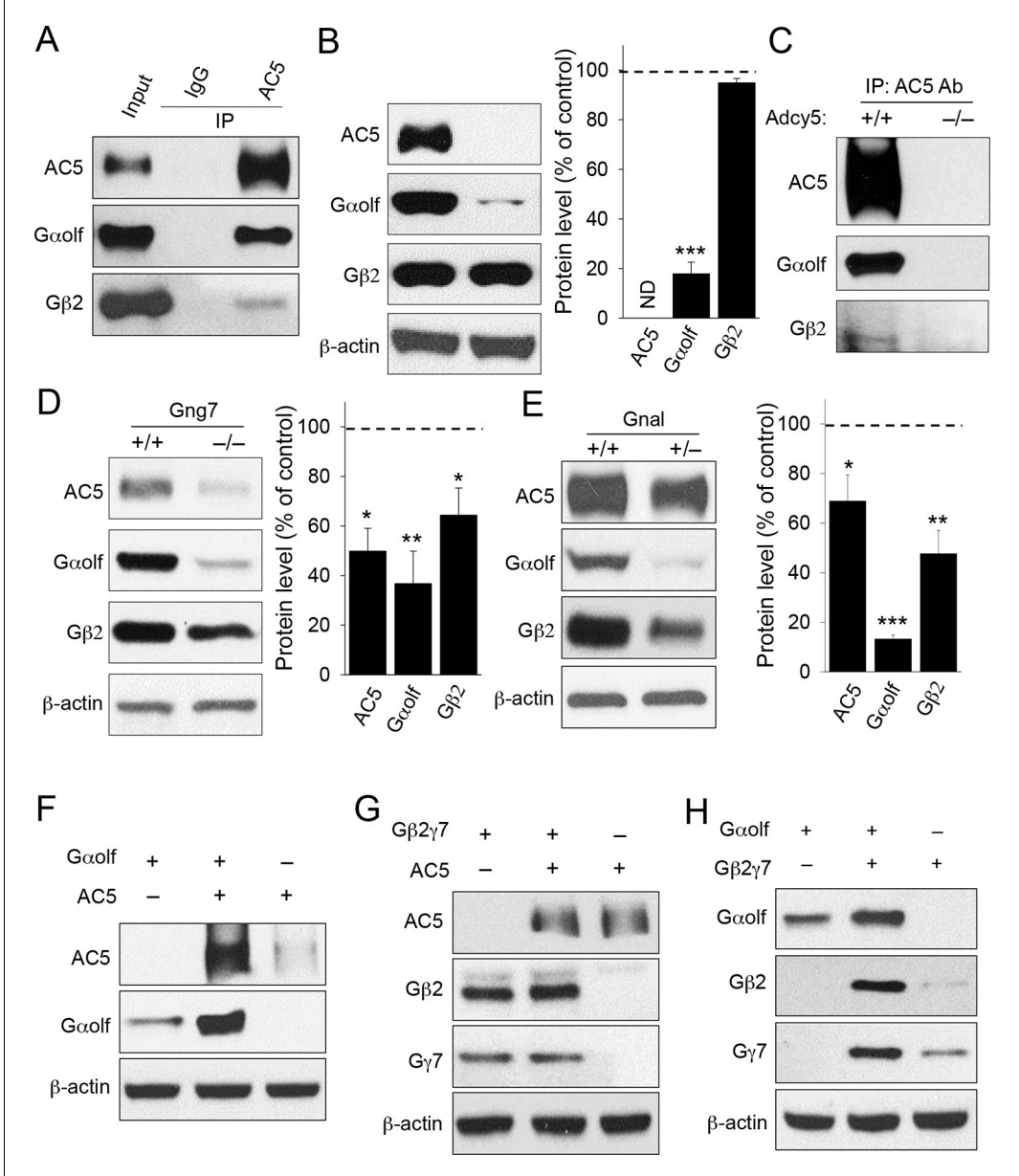

**Figure 1.** Co-dependence of heterotrimeric $G_{olf}$ subunits and AC5 in complex formation and expression. (**A**) Immunoprecipitation of AC5 complexes from striatal lysates of wild type mice. Specific anti-AC5 antibodies but not non-immune IgGs pull down $G\alpha_{olf}\beta_2\gamma_7$ subunits from native striatal tissues. (**B**) Significant reduction of $G\alpha_{olf}$ expression in striatum tissue from mice lacking AC5. Total striatal lysates were analyzed by immunoblotting using indicated antibodies and quantified by densitometry. ***$p<0.001$, Student's t-test, n = 3 mice. (**C**) Immunoprecipitation of AC5 from wild type (+/+) and Adcy5$^{-/-}$ tissues confirms specificity of AC5-$G_{olf}$ binding. Same anti-AC5 antibodies were used in both immunoprecipitaiton experiments and samples were processed in parallel. (**D**) Significant reduction of AC5 and $G\alpha_{olf}$ expression in striatal tissues from mice lacking $G\gamma_7$. Total striatal lysates from wild type and *Gng7$^{-/-}$* mice were analyzed by immunoblotting with indicated antibodies and quantified by densitometry. *$p<0.05$, **$p<0.01$, Student's t-test, n = 3 mice. (**E**) Significant reduction of AC5 and $G\beta_2$ expression in striatal tissues from mice with reduced expression of $G\alpha_{olf}$. Total striatal lysates from wild type and *Gnal$^{+/-}$* mice were analyzed by immunoblotting with indicated antibodies and quantified by densitometry. *$p<0.05$, **$p<0.01$, ***$p<0.001$, Student's t-test, n = 3 mice. (**F**) Mutual stabilization of AC5 and $G\alpha_{olf}$ upon co-transfection in HEK293 cells. Equal amounts of cDNAs were transfected into cells as described in the Methods section and protein expression was assessed by immunoblotting with specific antibodies (**G**) Lack of co-stabilization between AC5 and $G\beta_2\gamma_7$ subunits in co-transfected HEK293 cells. Indicated constructs were transfected into cells and changes in protein expression were monitored by immunoblotting with specific antibodies. (**H**) Mutual stabilization of $G\alpha_{olf}$ subunits with $G\beta_2\gamma_7$ subunits in co-transfected HEK293 cells. The experiment was conducted as described for panel G.

## PhLP1 facilitates the $G\alpha_{olf}\beta_2\gamma_7$ expression, assembly and functional activity.

In search of factors that may regulate the assembly of $G_{olf}$-AC5 complex, our attention was drawn to phosducin like protein 1 (PhLP1), which has been described to serve as a co-factor for the assembly of signaling complexes containing β subunits of heterotrimeric G proteins (*Willardson and Tracy, 2012*). In HEK293 cells, co-transfection with PhLP1 significantly increased the expression of all three $G_{olf}$ subunits (*Figure 2A*). In contrast, a dominant negative construct of PhLP1 (ΔNT-PhLP1), lacking the first 75 amino acids critical for its interaction with Gβ (*Lukov et al., 2005*) had no effect on the expression $G_{olf}$ subunits (*Figure 2A*). Consistent with the described impact of PhLP1 on Gβγ complexes (*Lukov et al., 2005*), co-expression with full-length PhLP1 resulted in a significant increase in the formation of the $G\beta_2\gamma_7$ complexes as evidenced by the fluorescence complementation assay with the split-Venus system (*Figure 2B*). In contrast, ΔNT-PhLP1 exerted the opposite effect and inhibited $G\beta_2\gamma_7$ assembly (*Figure 2B*). To examine the functional effects of PhLP1 in a context of the entire Golf heterotrimer, we tested the ability of $G\beta_2\gamma_7$ and $G\alpha_{olf}$ subunits to undergo a cycle of dissociation/re-association in response to changes in the D1R activity. We used a cell-based Bioluminescence Energy Transfer (BRET) assay to monitor the release of the $G\beta_2\gamma_7$ dimer in the presence of $G\alpha_{olf}$ (*Figure 2C*). In this assay, stimulation of D1R by dopamine results in the activation of $G\alpha_{olf}$, releasing Venus-tagged $G\beta_2\gamma_7$ to interact with the NanoLuc-tagged reporter, an event detected by changes in the BRET signal. Conversely, application of an antagonist SCH39166 facilitates complex re-association and quenching of the BRET response (*Figure 2D*). Reversing the sequence of ligand addition abolishes the response indicating that changes in the BRET signal are associated with the activation of the D1R receptors (*Figure 2D*). Furthermore, omitting D1R or $G_{olf}$ from the transfection also dramatically suppressed the signal indicating that the signal is specifically driven by the D1R-mediated activation of the $G_{olf}$ (*Figure 2E*). Using this assay we found that PhLP1 coexpression significantly increased dopamine-induced activation of $G\alpha_{olf}$-$G\beta_2\gamma_7$ heterotrimer (*Figure 2F,G*). In contrast, expression of the dominant negative ΔNT-PhLP1 significantly attenuated the BRET response (*Figure 2F,G*). These observations suggest that PhLP1 facilitates functional coupling of $G\alpha_{olf}G\beta_2\gamma_7$ to D1 receptors, likely owning to its ability to promote the assembly of $G\beta_2\gamma_7$ complexes.

## PhLP1 increases stimulatory regulation of AC5

The observations that AC5 expression is sensitive to changes in the levels of the Golf and that PhLP1 increases the expression of $G_{olf}$ subunits suggested that PhLP1 might modulate AC5 activity. We tested this possibility by measuring the effect of PhLP1 co-expression on AC5 expression. PhLP1 co-expression resulted in a significant increase (~1.7 fold) in AC5 expression (*Figure 3A*). This effect was paralleled by elevation of the basal cAMP levels (~4 fold) in the cells expressing AC5 (*Figure 3B*). In addition, AC5-containing cells showed enhanced cAMP generation in response to both the direct AC activator forskolin (~4 fold increase) and the β2-adrenergic agonist isoproterenol (~1.5 fold increase) when PhLP1 was overexpressed (*Figure 3B*). These effects were AC5-dependent as no stimulatory effect on cAMP generation was observed in the absence of AC5 (*Figure 3C*). Interestingly, while the increase in GPCR-driven cAMP production (likely through $G\alpha_s$) in the presence of PhLP1 closely matched the increase in the total AC5 levels, the effect of PhLP1 on both basal and forskolin-induced cAMP production was much larger. These observations suggest that PhLP1 may exert functional effects on cAMP production that are independent from increasing AC expression.

To gain mechanistic insight into the nature of this regulation in a more physiological setting, we performed adenylyl cyclase activity assays using membrane isolated from mouse striatum. Remarkably, purified recombinant full-length PhLP1 significantly enhanced basal AC activity (*Figure 3D*). This effect was not observed with ΔNT-PhLP1, suggesting that the effect likely involves an ability of PhLP1 to bind Gβγ. The reaction buffer in these assays did not contain GTP, therefore the effect of PhLP1 was unlikely to result from the classical activation of Gα. Given that in the striatal membranes AC5 forms a complex with $G\alpha_{olf}G\beta_2\gamma_7$ we hypothesized that PhLP1 may cause activation by scavenging Gβγ and releasing $G\alpha_{olf}$, making it available for the activation of AC5, as even GDP-bound free Gα subunits are capable of regulating AC (*Sunahara et al., 1997*). To test this notion, we compared the ability of forskolin and $G\alpha_s$-GTP to regulate AC activity in the striatal membranes in the absence or presence of PhLP1. $G\alpha_s$ is known to synergize with forskolin increasing its ability to regulate cyclase activity by promoting binding of forskolin to a high affinity site (*Dessauer et al., 1997*).

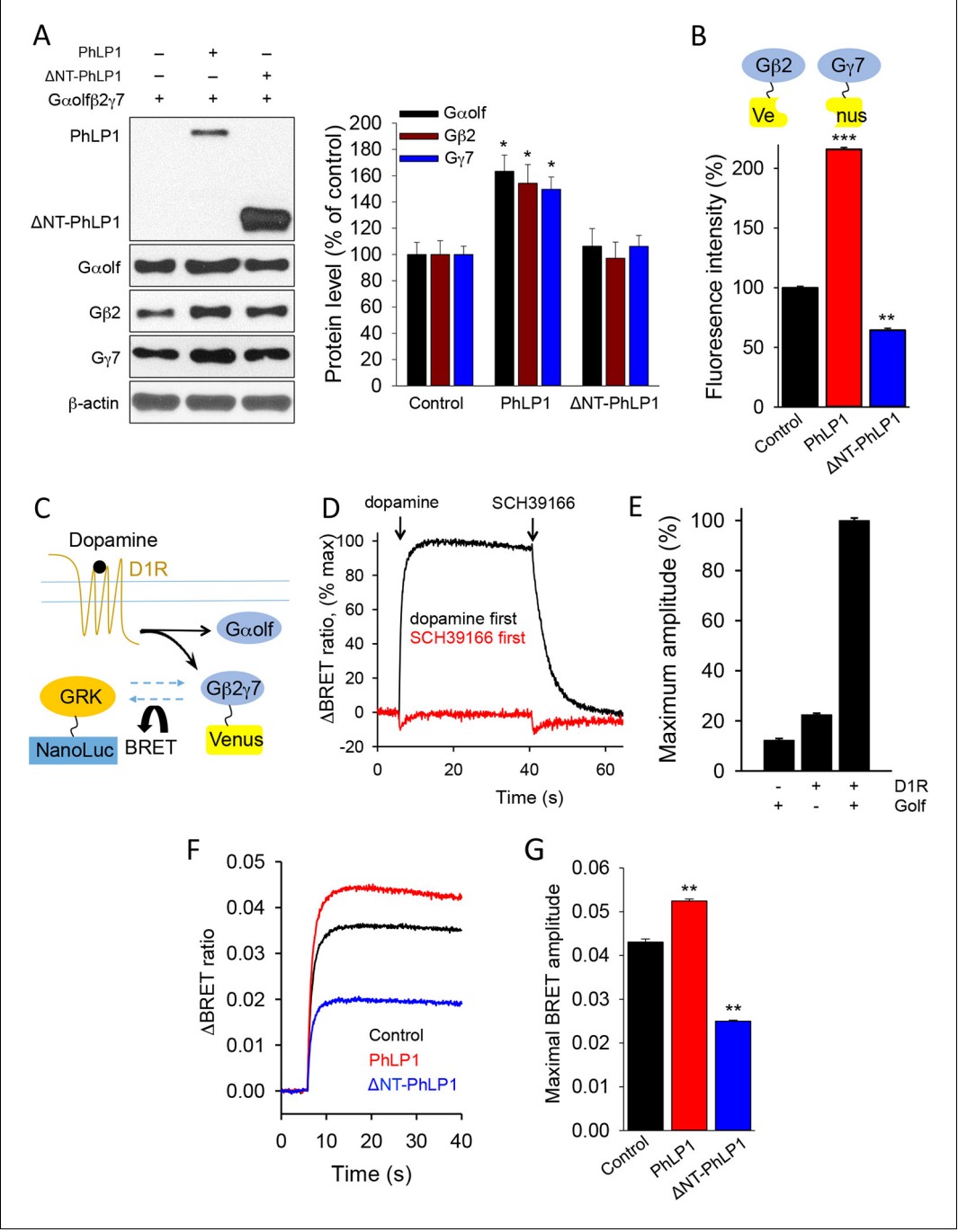

**Figure 2.** PhLP1 facilitates functional assembly of $G\alpha_{olf}\beta_2\gamma_7$ complex. (**A**) *Left,* Full length of PhLP1 but not its N terminally truncated mutant ΔNT-PhLP1 increases the expression level of $G\alpha_{olf}\beta_2\gamma_7$ subunits upon overexpression in HEK293 cells. *Right,* quantification of immunoblot data from 3 independent experiments. Data were normalized to the individual protein expression in the control group without PhLP1 transfection. Data were analyzed by One-Way ANOVA ($G\alpha_{olf}$ F[2, 9] = 8.731, p = 0.008; $G\beta_2$ F[2, 9] = 6.688, p = 0.017; $G\gamma_7$ F[2, 9] = 11.107, p = 0.004). *p<0.01 compared to the control group post hoc Tukey's test. (**B**) Full length PhLP1 facilitates, while ΔNT-PhLP1 inhibits $G\beta_2\gamma_7$ assembly. Venus fluorescence intensity was used as a readout of $G\beta_2\gamma_7$ complex assembly in a complementation experiment in transfected HEK293 cells. Data were analyzed by One-Way ANOVA (F[2, 15] = 2719.521, p<0.001). **p<0.01, ***p<0.001 compared with the control group, post hoc Tukey's test. (**C**) Schematic diagram of BRET sensor strategy for examining the dissociation and reassociation of $G\alpha_{olf}$ and $G\beta_2\gamma_7$ subunits upon D1Rs activation and inactivation. (**D**) Representative BRET response traces. Cells transfected with D1R and $G_{olf}$ were stimulated by 100 μM dopamine followed by 100 μM SCH39166 (black) or by 100 μM SCH39166

*Figure 2. continued on next page*

*Figure 2. Continued*

followed by 100 µM dopamine (red). First and second ligands were applied at 5 and 40 s, respectively. (**E**) Control experiments examining the requirement of both Golf and D1R to transduce the signal. Cells were transfected with the three different conditions, Golf only, D1R only, or D1R plus $G_{olf}$. Each bar represents the mean of 6 replicates. (**F**) Representative BRET signal traces in response to D1 receptor activation with dopamine (100 µM). (**G**) Comparison of maximal BRET ratios. Data were analyzed by One-Way ANOVA (F[2, 9] = 706.655, p<0.001). **p<0.01 compared with the control group, post hoc Tukey's test.

Consistent with our model, PhLP1 significantly enhanced forskolin-stimulated AC activity in striatal membranes and produced an approximately three-fold reduction in $EC_{50}$ for forskolin (***Figure 3E and F***). In contrast, PhLP1 failed to exert an effect on AC activated by adding exogenous $G\alpha_s$-GTPγS (***Figure 3E and G***) supporting the idea that PhLP1 activates AC5 through a Gα-dependent mechanism. In this case, added $G\alpha_s$-GTPγS likely out competed endogenous $G\alpha_{olf}$-GDP activating AC5 after being released by PhLP1. Thus, PhLP1 appears to regulate AC5 activity by a dual mechanism. On the one hand it enhances the expression of the AC5 by facilitating its complex formation with $G_{olf}$. On the other, it may act in promoting the receptor-independent release of Gαolf that stimulates AC5 activity.

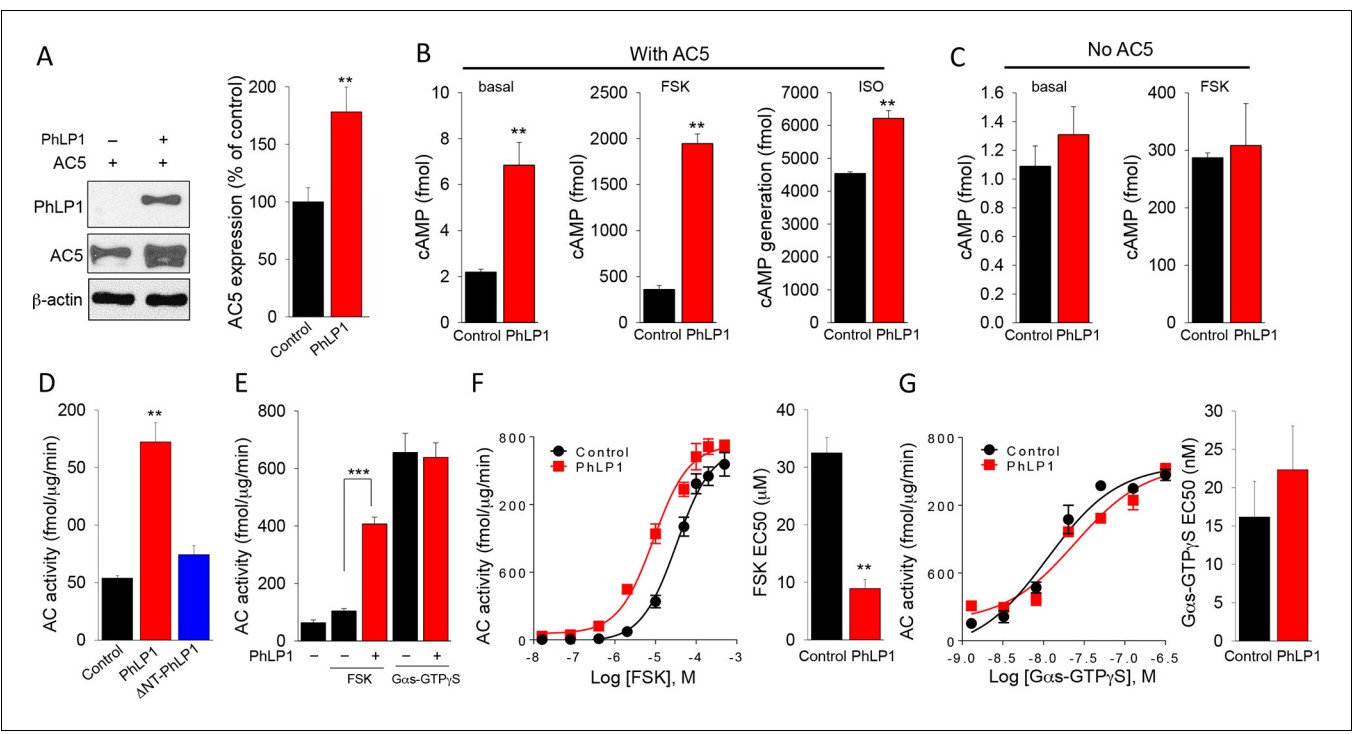

**Figure 3.** PhLP1 augments expression and activity of AC5. (**A**) Co-expression of PhLP1 significantly increases AC5 expression in HEK293 cells. **p<0.01 Students' t-test, n = 3. (**B**) PhLP1 enhances cAMP levels under basal condition, or in response to stimulation with forskolin (FSK, 1 µM, 5 min) or isoproterenol (ISO, 1 µM, 5 min) in AC5 expressing HEK293 cells. **p<0.01 Student's t-test, n = 3. (**C**) PhLP1 has no effect on cAMP generation in HEK293 cells without AC5 overexpression. n = 3 (**D**) Effects of purified recombinant full length PhLP1 and ΔNT-PhLP1 on adenylyl cyclase activity in striatal membranes. Assays were performed in the absence of GTP. Striatal membranes were pre-incubated with 0.5 µM of purified proteins at 4°C for 20 min and then subjected to adenylyl cyclase activity assay. Data were analyzed by One-Way ANOVA (F[2, 9] = 35.477, p<0.001). **p<0.01, post hoc Tukey's test. n = 3 (**E**) Purified recombinant PhLP1 enhances forskolin-stimulated adenylyl cyclase activity in striatal membranes. Membranes were pre-incubated with 0.5 µM of purified proteins at 4°C for 20 min. Membranes were then stimulated with 1 µM forskolin or 80 nM $G\alpha_s$-GTPγS. Data were analyzed by One-Way ANOVA (F(4, 15) = 52.291, p<0.001).***p<0.001, post hoc Tukey's test. n = 3 (**F**) Effect of purified recombinant PhLP1 on the dose response of forskolin-mediated activation of adenylyl cyclase in striatal membranes. The $EC_{50}$ for forskolin was 32.5 ± 2.7 µM in the control reaction and 8.9 ± 1.6 µM in PhLP1-treated reaction. **p<0.01 Student's t-test, n = 3. (**G**) Effect of purified recombinant PhLP1 on the dose response of $G\alpha_s$-GTPγS-mediated activation of adenylyl cyclase in striatal membranes. The $EC_{50}$ for $G\alpha_s$-GTPγS was 16.2 ± 4.7 nM in the control reaction and 22.3 ± 5.7 nM in PhLP1-treated reaction (n = 3 reactions).

## Elimination of PhLP1 does not impact survival and connectivity of striatal neurons

We next sought to examine the effect of PhLP1 on striatal signaling and physiology in vivo. In agreement with previous reports (*Schroder and Lohse, 2000*), we found PhLP1 protein to be abundantly expressed in the mouse striatum by immunoblotting (*Figure 4A*). To study the role of PhLP1 in the striatum, we ablated its expression selectively in the striatum by crossing the conditional PhLP1 mouse strain (*Pdcl$^{flx/flx}$*) with the striatal driver mouse line *Rgs9-Cre*, generating a conditional, striatal-specific elimination of PhLP1 (*Pdcl* cKO) mouse (*Figure 4B*). In this line recombination likely occurs postnatally, as expression of the *Rgs9* gene is induced around P3 to P6 (*Anderson et al., 2007*). We began our analysis by assessing possible anatomical changes because previous studies indicated that elimination of PhLP1 may lead to neuronal degeneration (*Lai et al., 2013*). Overall, striatal morphology of *Pdcl* cKO mice looked normal with no signs of degeneration at least until 3–4 months of age (*Figure 4C*). Morphometric analysis revealed a decrease in the total volume of striatal tissue in *Pdcl* cKO mice (*Figure 4D*). Counting the number of neurons (diameter > 5 μm) in the striatum tissue using Nissl staining revealed a significantly greater number of neurons in the *Pdcl* cKO mice. A greater number of cells together with a smaller volume that they occupy indicate that the sizes of individual striatal neurons are likely smaller. These changes are consistent with retarded maturation of neurons, a process controlled by the cAMP signaling (*Fujioka et al., 2004*; *Nakagawa et al., 2002*). We next analyzed projections of striatal medium spiny neurons to the target regions Globus Pallidus (GPe) and Substantia Nigra (SNr), revealed by immunostaining for enkephalin and substance P, respectively. Quantification of fluorescence intensity revealed no difference in the intensities of the signals for these markers, which were found in appropriate target areas (*Figure 4E*). In summary, these data indicate that loss of PhLP1 in the striatum does not lead to neuronal degeneration, but rather promotes neuronal survival while inhibiting their growth.

## Elimination of PhLP1 in striatal neurons destabilizes AC5-G$_{olf}$ complexes and leads to cAMP signaling deficits

Our studies in vitro and in heterologous expression system indicate a role for PhLP1 in the assembly of the G$\alpha_{olf}$G$\beta_2\gamma_7$ complex. Previous in vivo studies also demonstrated that PhLP1 is required for the assembly of G$\beta_1$ with G$\alpha$t$_1$ and G$\beta_3$ with G$\alpha_{t2}$ as well as G$\beta_5$ complexes with RGS9-1 (*Lai et al., 2013*, *Tracy et al., 2015*). Therefore, we proceeded to investigate the consequences of PhLP1 ablation on the expression of various subunits of heterotrimeric G proteins, RGS proteins and AC5 in the striatum (*Figure 5A*). Immunoblotting shows that the levels of PhLP1 protein were reduced by ~60% in *Pdcl* cKO striatum. Consistent with the results in transfected cells, we found that the levels of G$\alpha_{olf}$, G$\beta_2$ and AC5 were severely reduced in *Pdcl* cKO as well. Deletion of PhLP1 also had a detrimental effect on the expression of G$\beta_5$ and RGS9-2, as may have been expected from the studies on the rod and cone photoreceptors (*Lai et al., 2013*; *Tracy et al., 2015*). Interestingly, the effect was clearly selective as deletion of PhLP1 did not affect the expression of G$\beta_1$ and G$\alpha$ subunits possibly associated with it: G$\alpha_o$, G$\alpha_i$, G$\alpha_q$ (*Figure 5B*). Furthermore, the levels of another G$\beta_5$ associated protein, RGS7 were also unaffected. Analysis of the mRNA levels for corresponding down-regulated proteins showed no changes in the transcript levels, suggesting that PhLP1 likely contributes to protein stability rather than affects the expression through a transcriptional mechanism (*Figure 5C*). Therefore, it appears that PhLP1 selectively affects biosynthesis and/or assembly of the AC5 signaling complex that in addition to G$\alpha_{olf}$G$\beta_2\gamma_7$ also contains RGS9-2/G$\beta_5$ (*Xie et al., 2012*).

In agreement with the changes in protein levels, we found marked deficits in cAMP signaling. Basal cAMP levels were substantially lower in the striatum tissues from *Pdcl* cKO mice as compared to their control littermates (*Figure 5D*). Adenylyl cyclase activity measured with membranes isolated from the striatum was also lower upon the deletion of PhLP1 under both basal and forskolin-stimulated conditions (*Figure 5E*). Finally, the efficiency of D1R and A2AR coupling to cAMP production was lower in *Pdcl* cKO striatal membrane (*Figure 5E*). Together, these data indicate that in the striatum PhLP1 is necessary for high-level expression of the AC5 complex components and its functional activity.

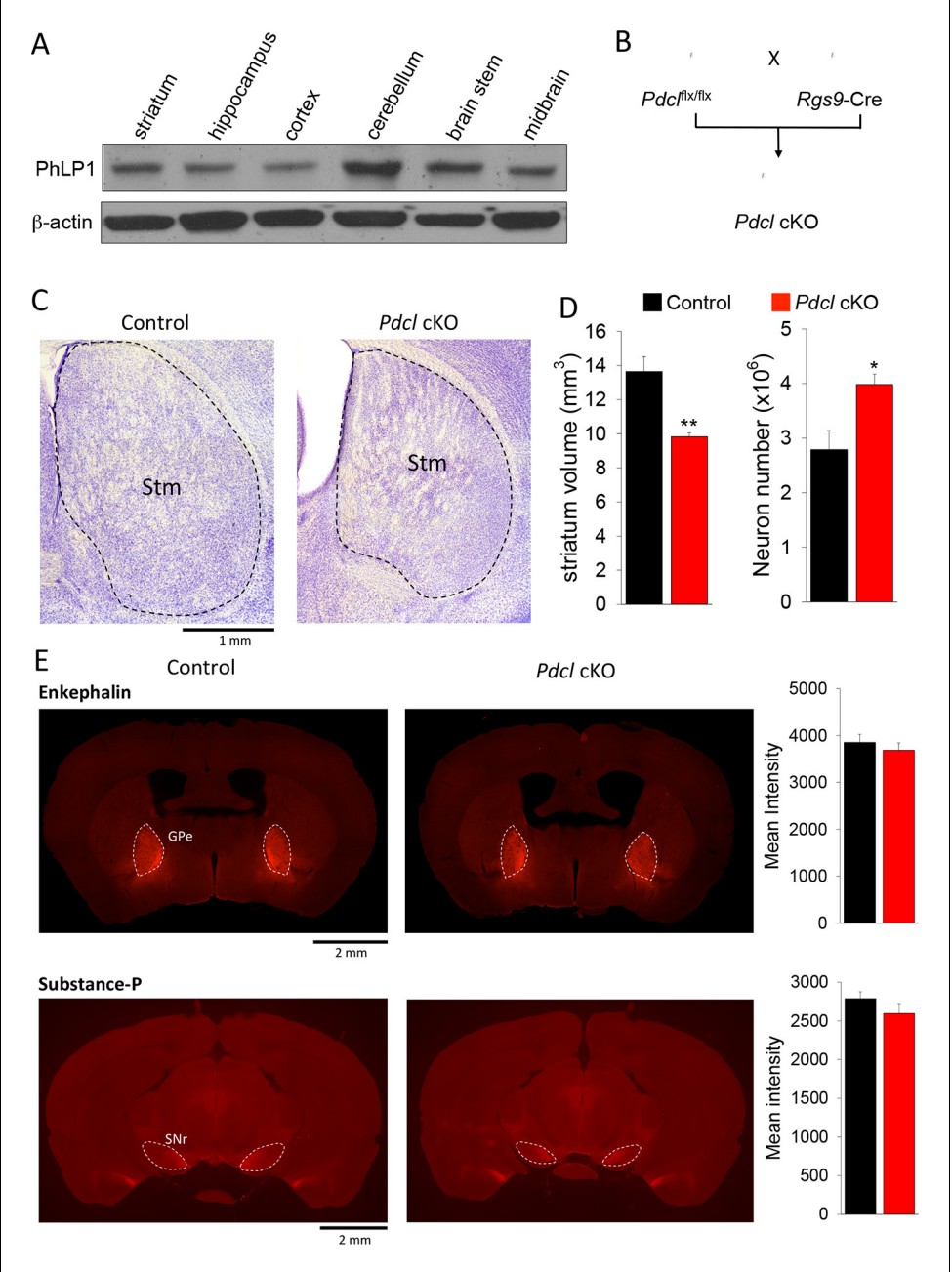

**Figure 4.** Elimination of PhLP1 does not impact survival and connectivity of striatal neurons. (**A**) PhLP1 expression in different brain regions from adult mice as determined by immunoblot analysis. (**B**) Generation of PhLP1 conditional knockout out. *Pdcl*<sup>flx/flx</sup> mice were crossed with *Rgs9-Cre* mice to generate striatal specific PhLP1 conditional knockout (*Pdcl* cKO) mice (**C**) Representative images of Nissl-stained coronal brain sections from adult control and *Pdcl* cKO mice. Stm, striatum. (Scale bar, 1 mm). (**D**) *Left*: Striatal volume of adult *Pdcl* cKO mice was reduced by 30% as compared with control mice (n = 4 mice each). Error bars represent SEM. Student's t test: \*\*p<0.01. *Right*: Counts of striatal neurons in *Pdcl* cKO and control mice obtained from Nissl-stained sections. The striatal neuron counts were increased by 43% compared with control mice (n = 4 mice each). Error bars represent SEM. Student's t test: \*p<0.05. (**E**) *Left*: Representative images of anti-enkephalin and anti-substance-P stained brain sections. GPe, external globus pallidus. SNr, substantia nigra pars reticulate. (Scale bar, 2 mm.) *Right*: Quantification of immunofluorescent signal for enkephalin and substance-P. Mean intensity of signals from both antibodies showed no significant difference between control and *Pdcl* cKO mice. Error bars represent SEM.

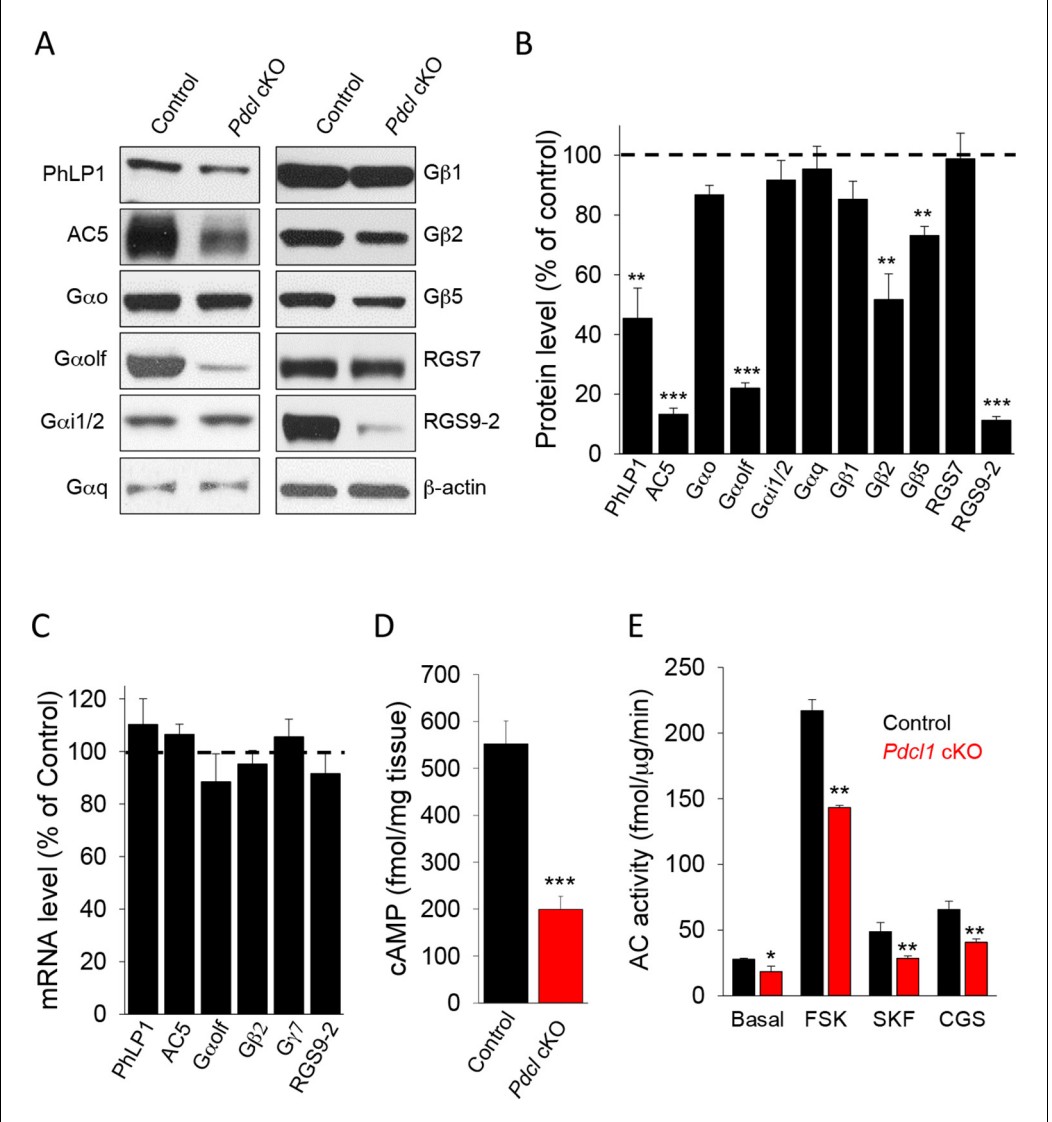

**Figure 5.** Elimination of PhLP1 in striatal neurons significantly impairs expression and function of AC5-Golf complex. (**A**) Representative immunoblot data of AC5, different G protein subunits and RGS proteins in stratal tissues from adult control or *Pdcl* cKO mice. (**B**) Quantification of protein levels. Data were normalized to the percentage of protein level in control mice. **p<0.01, ***p<0.001, Student's t-test, n = 3 mice. (**C**) mRNA quantification in striatum tissue from control and *Pdcl* cKO mice. Data were normalized to the percentage of mRNA level in control mice. (**D**) Basal cAMP level was reduced in the striatum tissue of *Pdcl* cKO mice. ***p<0.001, Students' t-test, n = 3 mice. (**E**) Striatal membrane adenylyl cyclase activity is reduced in *Pdcl* cKO mice. Adenylyl cyclase activity was measured either under basal conditions (without stimulation), or in the presence of forskolin (FSK, 1 μM), D1R specific agonist SKF38391 (SKF, 10 μM) or A2AR agonist CGS21680 (CGS, 10 μM). *p<0.05, **p<0.01, Student's t-test, n = 3 reactions.

## Elimination of PhLP1 in striatal neurons results in motor deficits and impaired psychomotor responses to drugs

To better understand the relevance of the observed molecular changes to normal physiology and pathology, we analyzed the behavioral consequences of eliminating PhLP1 in the striatum. We started by assessing the performance of *Pdcl* cKO mice in a battery of striatum-dependent tasks comparing their behavior to control littermates. In the open field test, *Pdcl* cKO mice exhibited normal habituation to a novel environment (*Figure 6A*) and had unaltered levels of basal locomotor activity as evidenced by both total distance traveled during the task (*Figure 6B*) and average

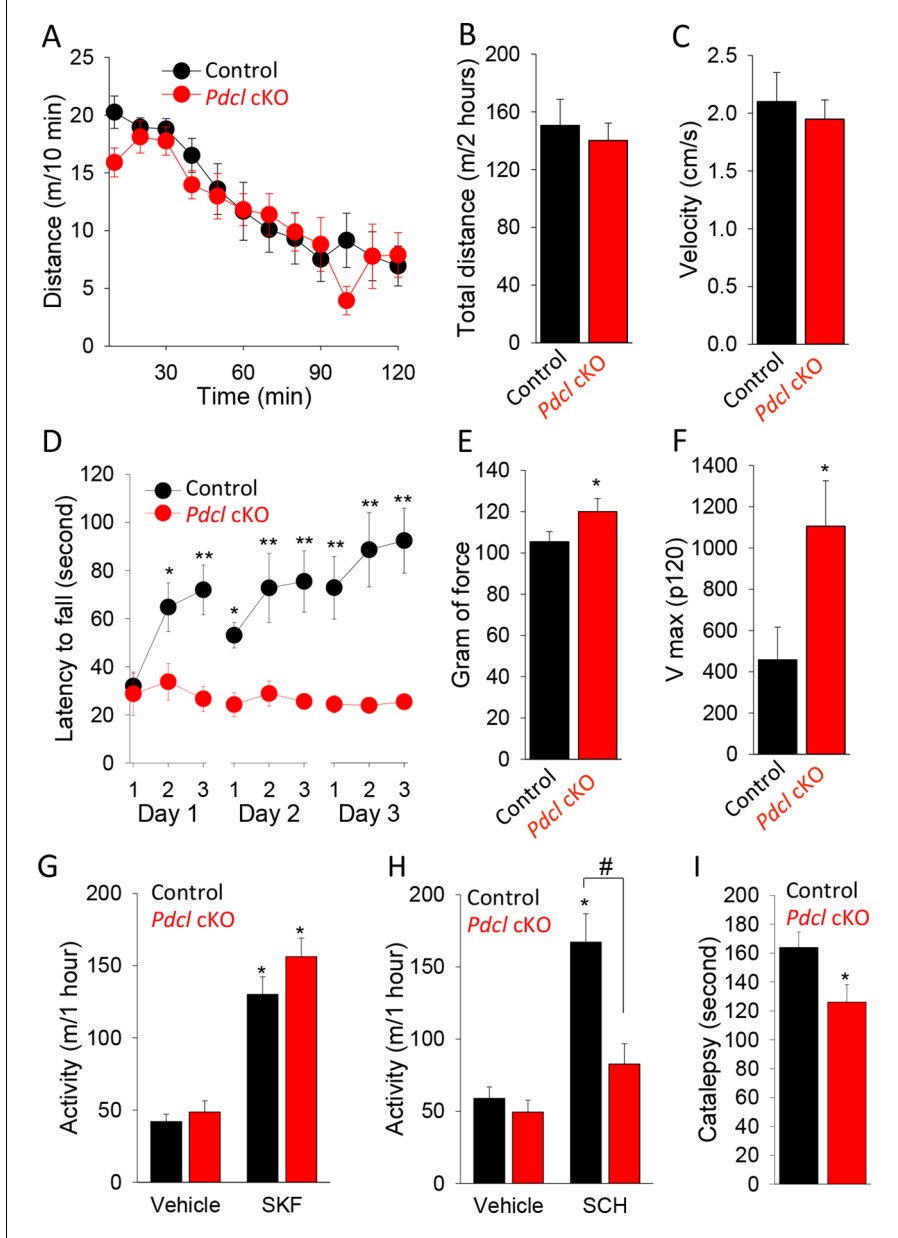

**Figure 6.** Behavioral consequences of PhLP1 elimination in striatal neurons. (**A**) *Pdcl* cKO mice display normal basal locomotion and habituation to a novel environment. (**B**) Total distance traveled in 2 hr in the open field chamber. (**C**) Average velocity in the open field chamber. (**D**) *Pdcl* cKO mice exhibit severe deficits in motor learning behavior in rotarod test. There were statistically significant differences between genotype as determined by Two-way ANOVA (F[1, 153] = 63.518, p<0.001). *p<0.05, **p<0.01, ***p<0.001, post hoc Tukey's test, n = 9 and 10 for control mice and *Pdcl* cKO mice, respectively. (**E**) Increased grip strength of the forelimbs in *Pdcl* cKO mice. *p<0.05, Student's t-test, n = 8 mice for each genotype. (**F**) Increased acoustic startle response of *Pdcl* cKO as measured by Vmax in response to 120 dB white noise bursts. (**G**) Normal locomotor response to D1R agonist SKF 38,393 (SKF, 50 mg/kg, i.p.) in *Pdcl* cKOcompared to control mice. Mice were injected with vehicle or SKF 38,393 (SKF, 50 mg/kg, i.p.) and immediately put in open field chambers. The locomotion was recorded for 1 hr. Data were analyzed by Two-way ANOVA (treatment F[1, 28]=96.068, p<0.001, genotype F[1, 28] = 2.679, p = 0.113). *p<0.01 post hoc Tukey's test compared to the vehicle control of the same genotype. n=8 mice per each genotype. (**H**) Blunted response to A2AR antagonist SCH58261 (SCH, 3 mg/kg, i.p.) treatment. Data were analyzed by Two-way ANOVA (treatment F(1, 28) = 42.819, p<0.001, genotype F(1, 28) = 20.181, p<0.001). *p<0.01 post hoc Tukey's test compared to the vehicle control of the same genotype, #p<0.05 post hoc Tukey's test in comparison between genotypes in response to SCH 58261. n = 8 mice per each genotype. (**I**) Reduced catalepsy in response

*Figure 6. continued on next page*

*Figure 6. Continued*

to D2R antagonist haloperidol (2 mg/kg, i.p.) in *Pdcl* cKO mice. Catalepsy was measured in the bar test 1 hr after haloperidol (2 mg/kg, i.p.) administration. *p<0.05, Student's t-test. n = 8 mice per each genotype.

locomotion velocity (*Figure 6C*). The *Pdcl* cKO mice also had normal thigmotaxis, stereotypy, pre-pulse inhibition and showed no signs of abnormal involuntary movements or clasping behavior typically associated with gross striatal dysfunction (data not shown).

We next tested animal motor behavior. In the accelerating rotarod task, mice of both genotypes that were naïve to the procedure performed equally during initial trial, indicating their normal motor coordination (*Figure 6D*). However, the behavior of *Pdcl* cKO mice was dramatically different when multiple trials were conducted. While the performance of control littermates improved with each subsequent trial, *Pdcl* cKO mice performed at the same level during all trial sessions (*Figure 6D*). These data indicate that knockout of *Pdcl* in the striatum completely abolishes motor learning in mice. *Pdcl* cKO mice also showed enhanced forelimb grips strength (*Figure 6E*) and augmented response to acoustic startle (*Figure 6F*), suggesting further deficits in processing sensorimotor stimuli.

To dissect the influence of PhLP1 on receptor-mediated neurotransmission, we performed pharmacological studies at the behavioral level. Injection of mice with the D1R selective agonist SKF38393 that activates striatal neurons of the direct pathway caused equal psychomotor activation in *Pdcl* cKO mice and their wild-type littermates (*Figure 6G*). In contrast, administration of A2AR antagonist SCH58261, that inhibits striatal neurons of the indirect pathway, had no effect on the behavior of *Pdcl* cKO mice while significantly increasing locomotor activity of wild-type littermates (*Figure 6H*). Conversely, the D2R antagonist haloperidol, that activates striatopallidal (indirect) pathway, induced catalepsy to a significantly greater extent in control littermates than in *Pdcl* cKO mice (*Figure 6I*). In summary, the behavioral data suggest normal baseline behavior of *Pdcl* cKO mice with selective deficits in the striatal-mediated motor learning and sensorimotor coordination, preferentially affecting the function of the striatopallidal pathway.

## Discussion

### Pre-assembly of stable signaling complexes between heterotrimeric $G\alpha_{olf}\beta_2\gamma_7$ and AC5

In this study we demonstrate that AC5, the key cAMP-producing enzyme in the striatum, forms stable complexes with its regulatory G protein species $G\alpha_{olf}\beta_2\gamma_7$ in vivo. We find that this interaction occurs at the basal state and can be detected without agonist application, which is typically required to promote dissociation of heterotrimers into $G\alpha$-GTP and $G\beta\gamma$ subunits making them competent for binding to effectors. We further demonstrate that binding to $G\alpha_{olf}\beta_2\gamma_7$ complex is required for the proteolytic stabilization of AC5 that results in its high expression level in the striatum. In the course of this study, we further confirmed previously reported interdependence among subunits of the $G_{olf}$ heterotrimer in which interactions between $G\alpha_{olf}$ and $G\beta_2\gamma_7$ are required for mutual proteolytic stabilization of the complex (*Iwamoto et al., 2004*; *Schwindinger et al., 2003*; *Schwindinger et al., 2010*). These observations are similar to noted interdependence of $G\alpha_{t1}\beta_1\gamma_1$ and $G\alpha_o\beta_3\gamma_{13}$ subunits observed in rod photoreceptors (*Lai et al., 2013*) and ON-bipolar cells (*Dhingra et al., 2012*) of the retina, respectively, and likely reflect a general principle in setting subunit stoichiometry in G protein heterotrimeric complexes. While there are several examples for the association of both $G\alpha$ and $G\beta\gamma$ subunits with effector molecule, e.g. PLC$\beta$, GIRK channels, AC isoforms (*Kovoor and Lester, 2002*; *Lyon et al., 2014*; *Sunahara et al., 1996*), to the best of our knowledge this study is the first to document a case in which stable association with all subunits of G protein heterotrimer is required for the stability of an effector. Taken together with biochemical studies showing that AC5 is capable of scaffolding inactive G protein heterotrimers (*Sadana et al., 2009*), our results suggest that in striatal neurons heterotrimeric $G_{olf}$-AC5 is assembled in a pre-coupled 'signalosome' in which subunits rearrange rather than physically dissociate upon GPCR activation.

## Role of PhLP1 in $G_{olf}$-AC5 complex assembly

We report that the chaperone protein PhLP1 plays a critical role in the assembly of the Golf-AC5 complex in striatal neurons. Previous studies have demonstrated the role of PhLP1 in the assembly of the complexes involving Gβ subunits of heterotrimeric G proteins. It is thought to function as a co-chaperone with the cytosolic chaperonin complex (CCT), assisting in retrieval of the Gβ subunits emerging from the CCT and presenting them for the association with the Gg subunits (*Willardson and Tracy, 2012*). In addition to assisting folding of conventional Gβγ complexes, PhLP1 chaperones the formation of the structurally similar complexes involving the atypical $Gβ_5$ subunits and the Gγ-like domains in RGS proteins (*Howlett et al., 2009*). Consistent with this model, deletion of PhLP1 in photoreceptors disrupts the formation of $Gβ_1γ_1$ (in rods), $Gβ_3γ_8$ (in cones) and $Gβ_{5L}$/RGS9-1 complexes (in both rods and cones), dramatically decreasing their expression (*Lai et al., 2013*; *Tracy et al., 2015*). Similarly, overexpression of the dominant negative mutant of PhLP1 deficient in Gβ binding in rods down-regulates the expression of the $Gβ_1γ_1$ and $Gβ_{5L}$/RGS9-1 complexes (*Posokhova et al., 2011*). Interestingly, loss of PhLP1 function also results in a dramatic decrease in the expression of $Gα_{t1}$ and $Gα_{t2}$ subunits in rods and cones, respectively (*Lai et al., 2013*; *Tracy et al., 2015*). Given that stability of these Gα subunits depends on their complex formation with Gbg (*Kolesnikov et al., 2011*; *Lobanova et al., 2008*; *Nikonov et al., 2013*), the role of PhLP1 in stabilizing $Gα_{t1}$ and $Gα_{t2}$ is likely indirect and stems from its effects on Gβγ complex assembly.

Similar to the situation in photoreceptors, we observe that knockout of PhLP1 in striatal neurons results in marked down-regulation in the expression of all subunits of heterotrimeric $G_{olf}$ complex. Direct side-by-side comparison of the impact on protein expression produced by elimination of individual subunits of the $G_{olf}$-AC5 complex affords a unique opportunity to analyze the reciprocal relationship between complex components. Our results suggest a hierarchical relationship in the assembly of the complex. The stability of Gαolf requires its association with both $Gβ_2γ_7$ and AC5, but the stability of $Gβ_2γ_7$ depends only on the association with Gαolf and not AC5. In turn, stability of AC5 is dependent on both $Gβ_2γ_7$ and $Gα_{olf}$. Given that PhLP1 directly binds to Gbg subunits (*Savage et al., 2000b*; *Thibault et al., 1997*) and has been shown to be required for folding $Gβ_2γ_7$ in transfected cells (*Howlett et al., 2009*), we propose the following model for the role in PhLP1 the assembly of the $G_{olf}$-AC5 complex. PhLP1 assists the assembly of the $Gβ_2γ_7$ complex, increasing its expression, which in turn upregulates $Gα_{olf}$. Higher levels of $Gα_{olf}β_2γ_7$ promote stabilization of the AC5 by forming pre-coupled complexes with it. Thus, PhLP1 triggers a chain of events resulting in the stabilization of the entire AC5-$G_{olf}$ complex ensuring its high expression level and setting the stoichiometry (*Figure 7*).

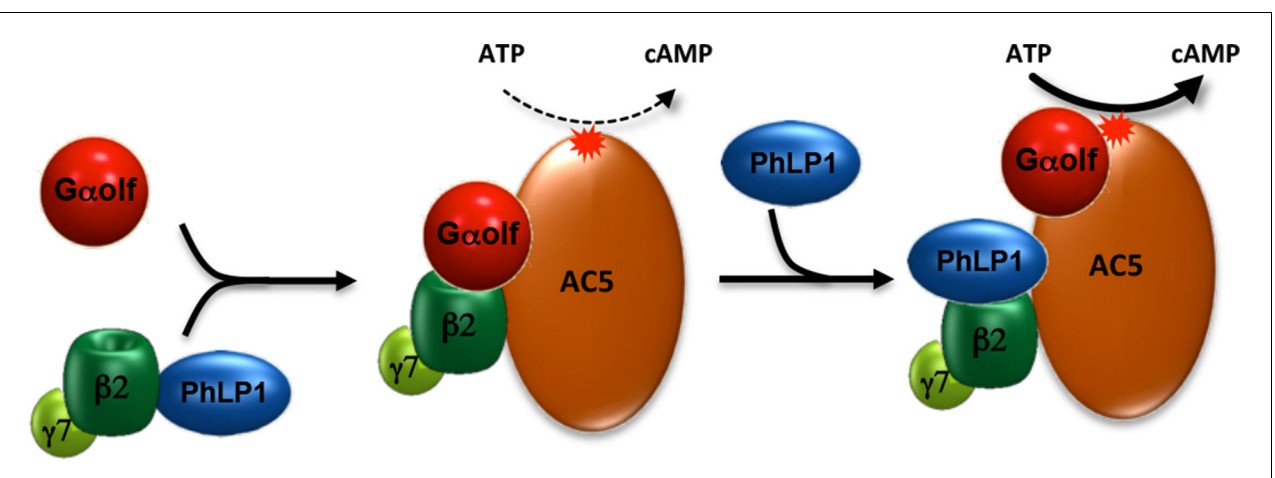

**Figure 7.** Schematic illustration of PhLP1 involvement in regulating $G_{olf}$-AC5 complex assembly and signaling. PhLP1 promotes biogenesis of $Gβ2γ7$ and assembly of its complex with $G_{olf}$. The trimeric $Gα_{olf}β_2γ_7$ forms stable complexes with AC5 contributing to its proteolytic stability. In addition, PhLP1 regulates cAMP production by influencing $G_{olf}$ arrangement on AC5.

It is interesting that PhLP1 loss in striatal neurons affected G protein complexes in a selective fashion. Studies in transfected cells indicate that PhLP1 participates in folding and stabilization of virtually all Gβγ combinations including complexes of Gβ$_5$ with RGS9 and RGS7 (*Howlett et al., 2009*). Yet, we find that in the striatum, PhLP1 elimination has no effect on the expression of many G protein subunits, including Gβ$_1$, a subunit clearly impacted by PhLP1 loss in rod photoreceptors (*Lai et al., 2013*). We propose that this apparent selectivity may be explained by higher susceptibility of G proteins with high expression levels to the destabilizing effects associated with PhLP1 loss than those expressed at moderate to low levels. In this scenario, PhLP1 action may be the rate-limiting factor required for achieving high expression of G proteins with abundant mRNA expression in a particular neuronal population, e.g. Gα$_{t1}$β$_1$γ$_1$, Gα$_{t2}$β$_3$γ$_8$ and Gαolfβ$_2$γ$_7$ overexpressed in rods, cones and striatal neurons, respectively. However, at this point, we also cannot rule out an alternative explanation that cellular heterogeneity in the striatum contributes to selectivity of the effects. Although *Rgs9-Cre* driver line that we used to delete PhLP1 is active in 95% of neurons in the striatum, it might have no effect on other cells such as glia that might contain a higher abundance of G protein subunits that we find to be unregulated by PhLP1.

## Role of PhLP1 in regulating neurotransmitter signaling to cAMP production

It is interesting to consider the results of this study in light of the mechanisms linking GPCR signaling in striatal neurons to cAMP production and behavior. The G$_{olf}$ heterotrimer is known to be essential for coupling both D1 and A2A receptors in direct and indirect striatal neurons to stimulation of cAMP production (*Herve, 2011*). Knockout of *Gnal* or *Gng7* that encode Gα$_{olf}$ or Gγ$_7$ respectively, markedly reduces the ability of D1R and A2AR agonists to increase cAMP, and this biochemical observation is paralleled by muted behavioral responses of mice to psychostimulants that activate these receptors (*Corvol et al., 2001*; *Schwindinger et al., 2003*; *Zhuang et al., 2000*). Acute psychomotor responses to drugs appear to be particularly sensitive to loss of G$_{olf}$, while adaptive behaviors to repeated drug exposure are preserved (*Corvol et al., 2007*; *Schwindinger et al., 2010*). Interestingly, levels of G$_{olf}$ complex dictate particular signaling outcomes upon receptor stimulation. For example, while Gα$_{olf}$ is critical for the ability of D1R to stimulate both cAMP production and ERK phosphorylation, Gα$_{olf}$ haploinsufficiency leads to selective deficits in cAMP signaling without detrimentally affecting coupling to ERK (*Corvol et al., 2007*). This signaling dichotomy becomes even more pronounced at the level of AC5. Although, activation of either D1R and A2A clearly results in the increase in cAMP levels (*Corvol et al., 2001*; *Iwamoto et al., 2003*; *Lee et al., 2002*) and AC5 mediates nearly 80% of this effect (*Lee et al., 2002*), elimination of AC5 in mice does not diminish behavioral responses of mice to the administration of A1R and D1R agonists (*Iwamoto et al., 2003*; *Lee et al., 2002*). At the same time, mice lacking AC5 have abolished responses to D2 antagonists, suggesting greater impact of signaling via AC5 in indirect pathway neurons (*Lee et al., 2002*). Balance of neurotransmitter signaling between direct and indirect pathway striatal neurons is thought to set a tightly orchestrated coordination of movements (*Graybiel, 2005*; *Nelson and Kreitzer, 2014*). In agreement with this idea, mice lacking AC5 or Gγ$_7$ show profound deficits in motor learning and coordination (*Iwamoto et al., 2003*; *Kheirbek et al., 2009*; *Sasaki et al., 2013*). Furthermore, mutations in genes encoding AC5 and Gα$_{olf}$ in humans cause dystonia, a disorder characterized by involuntary movements (*Carapito et al., 2014*; *Fuchs et al., 2012*; *Kumar et al., 2014*). Thus, it is likely that G$_{olf}$-AC5 axis is involved in more subtle coordination of signaling in striatal neurons that sometimes might not be evident from measuring gross motor responses to pharmacological treatment with receptor ligands.

Our findings with a mouse model lacking PhLP1 in the striatum agree well with the expectations based on the analysis of mice with disruptions in AC5 and G$_{olf}$ components and also provide additional insight that helps further clarify integration of neurotransmitter signaling in the striatum. Our *Pdcl* cKO model displays complete lack of motor learning, deficits in psychostimulatory effects of A2AR antagonism and diminished neuroleptic responses to D2R antagonism. As with studies on *Adcy5-/-* mice (*Herve, 2011*; *Lee et al., 2002*), perhaps the most surprising result of our studies is the intact responses of mice lacking PhLP1 to D1R agonism. This observation is particularly striking given the profound down-regulation of AC5, Gα$_{olf}$ and Gβ$_2$γ$_7$ in these mice paralleled by equal deficits in coupling of both D1R and A2AR to cAMP production. We think that this selectivity is likely related to downregulation of all signaling components rather than their complete loss. It is possible

that striatonigral and striatopallidal neurons have differential sensitivity to changes in the efficiency of the $G_{olf}$-AC5 coupling, creating signaling imbalance when the expression of signaling components is diminished. This effect could be either exacerbated or compensated by the loss of RGS9-2 that also controls AC5 activity (*Xie et al., 2012*) and signaling downstream from D2 receptors (*Cabrera-Vera et al., 2004*; *Celver et al., 2010*; *Rahman et al., 2003*), further contributing to signaling imbalance. Alternatively, preservation of D1R mediated behavioral responses in *Pdcl* cKO mice may also indicate that signaling pathways other than cAMP initiated by D1 receptors (*Nishi et al., 2011*) play compensatory role. In any event, our study demonstrates the importance of proper Golf-AC5 complex expression and assembly for the balance of the neurotransmitter signaling in striatal neurons and introduces PhLP1 as a critical regulator of the process and an interesting molecular player to consider in the pathology of dystonia and possibly other movement disorders.

## Materials and methods

### Mice and DNA constructs

*Pdcl* $^{flx/flx}$ mice (*Lai et al., 2013*) were crossed with *Rgs9-Cre* mice (*Dang et al., 2006*) to generate striatal specific PhLP1 conditional knockout (*Pdcl* cKO) mice. *Pdcl* $^{flx/flx}$ Cre(-) control littermates derived from heterozygous breeding pairs were used for all experiments. Mice were housed in groups on a 12 hr light–dark cycle with food and water available *ad libitum*. Males and females (2–5 months) were used for all experiments. All procedures were approved by the Institutional Animal Care and Use Committee (IACUC) at The Scripps Research Institute. Generation and characterization of *Gnal* $^{+/-}$ (*Belluscio et al., 1998*), *Gng7* $^{-/-}$ (*Sasaki et al., 2013*)) and *Adcy5* $^{-/-}$ (*Lee et al., 2002*) mice have been described previously.

Full length of PhLP1 and N-terminal (1–75 a.a.) truncation of PhLP1 (ΔNT-PhLP1) were cloned into pcDNA3.1 vector as previously (*Lukov et al., 2005*). Venus155-239-G$\beta_2$ and Venus1-155-G$\gamma_7$ constructs were generated by replacing G$\beta_1$ in Venus155-239-G$\beta_1$ construct with G$\beta_2$ and G$\gamma_2$ in Venus1-155-G$\gamma_2$ construct with G$\gamma_7$ (*Hollins et al., 2009*). Construction of masGRK3ct-Nluc (*Posokhova et al., 2013*) and Flag tagged AC5 (*Xie et al., 2012*) was reported previously. G$\alpha_{olf}$ and D1R cDNAs were purchased from Missouri S&T cDNA Resource Center. Flag-tagged Ric-8B in pcDNA3.1 (*Von Dannecker et al., 2006*) were gifts from Dr. Bettina Malnic.

### Antibodies, cell-lines, immunoprecipitation and immunoblotting

Hybridoma cell lines expressing mouse monoclonal AC5 antibody against human AC5 peptide CGN-QVSKEMKRMGFEDPKDKN were commercially generated by Genscript. Hybridomas were cultured in DMEM/F-12 supplemented with 10% fetal bovine serum and 1% penicillin/streptomycin. Synthetic peptide (CGNQVSKEMKRMGFEDPKDKN) was covalently immobilized to beaded agarose using SulfoLink Immobilization Kit (Pierce Biotechnology). Antibodies were purified from collected hybridoma culturing medium by affinity chromatography using immobilized antigen peptide. Other antibodies used were against: c-myc (Genescript), G$\beta_1$ (*Lee et al., 2004*) and G$\alpha_{olf}$ (*Corvol et al., 2001*). PhLP1 (*Lai et al., 2013*); β-actin (AC-15) (Sigma-Aldrich, St. Louis, MO); G$\beta_2$, G$\alpha_q$, G$\gamma_7$ and G$\alpha_o$ (K-20) (Santa Cruz Biotechnology, Dallas, TX); G$\alpha_{i1/2}$ (Affinity BioReagents, Golden, CO); GFP (clones 7.1 and 13.1; Roche Applied Science); G$\beta_5$ and RGS7 and RGS9-2 (*Martemyanov et al., 2005*).

For immunoblot analysis or immunoprecipitation, striatal tissue (~15 mg) or transfected HEK293T/17 cells were homogenized by sonication in lysis buffer (1xPBS, 150 mM NaCl, 0.5% dodecyl nonaoxyethylene ether (C12E9) containing complete protease inhibitor cocktail (Roche) and phosphatase inhibitor cocktail 1 and 2 (Sigma). The homogenate was centrifuged at 16,000×g for 15 min. For immunoprecipitation, the supernatant was incubated with 3 μg of antibody as indicated and 10 μL of protein G beads for 1 hr at 4°C. Beads were washed 3 times with lysis buffer. Protein sample was eluted in SDS sample buffer containing 4M urea, incubated at 42°C for 15 min, resolved by SDS-PAGE, transferred onto PVDF membrane and subjected to immunoblot analysis.

All experiments involving cultured cells were performed in HEK293/17 cell line obtained from ATCC (Manassas, VA). The company certifies authenticity of the cell line, and guarantees it to be free of contaminants and pathogens. The cells were maintained in standard DMEM medium, and frozen in aliquots from the stock received from the ATCC. The cells were grown for no more than 20

passages. The laboratory has tested tissue culture facility and found it to be free of mycoplasma contamination.

## Recombinant proteins and membrane preparations

Recombinant Gαs were expressed in BL21 (DE3) *E. coli* strain and purified by affinity chromatography on HisTALON column (Clontech, Mountain View, CA) as described previously (*Lee et al., 1994*). Gαs was activated by incubation with 20 µM GTPγS in the assay buffer containing 20 mM Tris-HCl pH 7.8, 10 mM $MgCl_2$, 1 mM EDTA, 1 mM dithiothreitol for 30 min at 30°C. The unbound GTPγS was then removed by Zeba spin desalting column (Life Technologies, Carlsbad, CA). His-tagged full-length PhLP1 as well as the N-terminal (1–75 a.a.) truncation of PhLP1 were purified from *E. coli* as previously described (*Savage et al., 2000a*). The purity of the recombinant proteins was assessed by Coomassie staining following gel separation and was found to be at least 80%.

For membrane preparation, striatal tissues were homogenized in buffer containing 250 mM sucrose, 20 mM Hepes pH 8.0, 1 mM EDTA, 2 mM $MgCl_2$, 1 mM DTT and proteinase inhibitors. The homogenate was centrifuged at 2000 g to remove nuclei, followed by centrifugation at 25,000 rpm in Beckman SW28.1 rotor for 35 min in 23/43% sucrose gradient to isolate the membrane fraction. The plasma membranes were carefully collected from the layer at the 23/43% sucrose interface. The protein concentrations in plasma membrane preparations were then determined by Pierce 660nm Protein Assay Reagent (Thermo Fisher Scientific, Waltham, MA).

## cAMP measurement and adenylyl cyclase activity assay

Striatal tissues were homogenized in 0.1 N HCl (20 µL per mg tissue). Lysates were centrifuged at 600 g for 10 min. Supernatants were collected, diluted 50-fold and cAMP concentrations were quantified using a cAMP enzyme immunoassay kit (cAMP Direct EIA) following the acetylated version protocol (Enzo Life Sciences, Farmingdale, NY). The activity of adenylyl cyclase in striatal membrane preparations (1 µg protein/reaction) was determined as described previously (*Xie et al., 2012*). Briefly, 1 µg of striatal membrane was treated with vehicle (basal), or indicated stimulator for 10 min at 30°C in adenylyl cyclase assay buffer (50 mM HEPES pH 8.0, 0.6 mM EDTA, 100 µg/mL BSA, 100 µM 3-isobutyl-1-methylxanthine (IBMX), 3 mM phosphoenolpyruvate potassium, 10 µg/mL pyruvate kinase, 5 mM $MgCl_2$ and 100 µM adenosine triphosphate (ATP). Reactions were stopped by adding an equal volume of 0.2 N HCl. For dose response curves of Gαs-GTPγS or forskolin (FSK) stimulated adenylyl cyclase activity experiments, striatal membranes were pre-incubated with 0.5 µM purified PhLP1 for 20 min on ice and then stimulated with increasing doses of Gαs-GTPγS or FSK as indicated. The resulting cAMP in the sample was determined by cAMP Direct EIA kit.

## mRNA quantification

Total RNA from striatal tissues was extracted and quantified as previously (*Orlandi et al., 2015*). Briefly, striatal tissues were homogenized in TRIZOL reagent (Invitrogen, ) according to the manufacturer's instructions. cDNA was generated from 0.5 µg of total RNA using qScript cDNA SuperMix (Quanta Biosciences, Gaithersburg, MD) according to the manufacturer's instructions. To analyze the RNA expression pattern of the target genes, the 7900HT Fast Real-Time PCR System (Applied Biosystems) was used with the Taqman gene expression master kit. Three biological replicates and four technical replicates for each sample were used. 10 ng of each sample were used in each real-time PCR (TaqMan Gene Expression Assay ID probes: *Pdcl*: Mm01327170_m1; *Adcy5*: Mm00674122_m1; *Gnal*: Mm01258217_m1; *Gnb2*: Mm00515865_g1; *Gng7*: Mm00515876_m1; *Rgs9*: Mm01250425_m1; Applied Biosystems). The expression ratio of the target genes was calculated using $2^{-\Delta\Delta C_T}$ method (*Livak and Schmittgen, 2001*) with 18S ribosomal RNA (ID: Mm03928990_g1) as reference. Data are shown as mean ± S.E.M.

## Bioluminescence resonance energy transfer measurements

Agonist-dependent cellular measurements of bioluminescence resonance energy transfer (BRET) between masGRK3ct-Nluc and Venus-Gβ₂γ₇ were performed to visualize the action of G protein signaling in living cells as previously described with slight modification (*Kumar et al., 2014*). Briefly, dopamine D1 receptor, Gαolf, Venus156-239-Gβ₂, Venus1-155-Gγ₇, Flag-Ric-8B and masGRK3ct-Nluc constructs together with PhLP1 or ΔNT-PhLP1 were transfected into HEK293T/17 cells at a

1:6:1:1:1:1:1 ratio using lipofectamine LTX transfection reagent (Invitrogen). 7.5 μg total DNA was delivered per $3.5 \times 10^6$ cells in a 6-cm-dish. 16–24 hr post transfection, cells were stimulated with 100 μM dopamine followed by treatment with 100 μM SCH39166. The BRET signal was determined by calculating the ratio of the light emitted by the Venus-Gβ2γ7 (535 nm) over the light emitted by the masGRK3ct-Nluc (475 nm). The average baseline value recorded prior to agonist stimulation was subtracted from BRET signal values, and the resulting difference (ΔBRET) was obtained.

## Histology

Mice were anesthetized by Avertin (tribromoethanol) and perfused transcardially with phosphate buffered saline (PBS) and 4% paraformaldehyde (PFA). Brains were collected and postfixed in 4% PFA overnight, and stored in 30% sucrose solution for cryoprotection for 3–5 days. Brains were sectioned into 50 μm slices by a sliding microtome (SM2000R, Leica). After sectioning, sample slices were stored in PBS at 4°C or in antifreeze solution at -20°C for long-term storage.

Immunofluorescent staining was performed as described previously (Gharami et al., 2008). Primary antibodies against enkephalin and substance-P (Immunostar Inc.; 1:1,1000) were used for detecting the projection from dopamine receptor D1 or D2 medium spiny neurons. Fluorescent dye-conjugated secondary antibodies (Alexa Fluor 594, Jackson ImmunoResearch Inc.) were applied during staining. The staining was performed on two brain sections including either external globus pallidus (GPe) or substantia nigra pars reticulata (SNr). After acquiring images by using a 4X objective (CFI Plan Apo, Nikon) on Nikon Ti microscope, the Nikon NIS-Elements software was used to measure mean intensity in designated area. Then values of control and mutant groups, with backgrounds subtracted, were evaluated by Student's unpaired t-test.

For neuronal cell counting in striatum, we first performed Nissl-staining in six coronal sections with 300 μm interval. Stereo Investigator software (MicroBrightField Inc.) was employed to measure the volume of striatum and evaluate the neuronal number. The number of neurons was estimated using a fractionator sampling method described previously (Baydyuk et al., 2011). The analysis was performed blinded to the genotypes.

## Behavioral studies

Locomotor activities were evaluated in automated video tracking ANY-maze open field chambers (Stoelting, Wood Dale, IL) under illuminated conditions. Mice were habituated to the testing room for 1 hr before the test on each day. On the first day, naïve mice were placed in the novel chambers without injection, and allowed to explore the chambers for 2 hr. Horizontal activity was measured in terms of the total distance traveled or distance traveled in 10-min intervals. Thigmotaxis (wall-hugging) for each subject was determined by dividing the distance traveled in the 10-cm-wide perimeter of the chamber by the total distance traveled during the 2-hr session. For pharmacological studies, mice were injected with vehicle (10 mL/kg, i.p.), D1 receptor agonist SKF 38,393 (50 mg/kg, i.p.), or A2a receptor antagonist SCH 58,261 (3 mg/kg, i.p.), and then immediately placed in the open-field chambers. Activity was monitored for 1 hr. Dose response of the same drug treatments were done on the same mice starting from the low dose first with 2-day intervals between different doses. SKF 38,393 was dissolved in saline, and SCH 58,261 was dissolved in saline solution containing 10% DMSO and 10% Kolliphor EL (Sigma Aldrich, St. Louis, MO).

Accelerating rotarod performance was tested using a five-station rotarod treadmill (IITC Rotarod, IITC Life Science, Woodland Hills, CA). Mice were habituated in the test room for an hour before the testing. Three trials were performed per day over 3 days for a total of nine trials for each animal. After placing a mouse on the rod, it was accelerated from 8 to 22 r.p.m. in 2 min. The endurance of mice on the rotarod was measured by the time to fall to the floor of the apparatus, or to turn around one full revolution while hanging onto the drum.

Catalepsy was measured in the bar test. Briefly, one hour after haloperidol (2 mg/kg, i.p.) administration, mice forepaws were gently placed over a horizontal bar fixed at a height of 5 cm above the working surface. The length of time during which the animal retained this position until the removal of one of its forepaws was recorded with the cutoff time of 180 s.

Grip strength of mice was assessed using Grip-Strength Meter (Ugo Basile, Italy). The mouse was placed over a base plate, in front of a grasping bar. The bar was fitted to a force transducer connected to the Peak Amplifier. When pulled by the tail, the mouse instinctively grasped to the bar

until the pulling force overcomes their grip strength. After the mouse lost its grip on the grasping bar, a peak preamplifier automatically stored the peak pull force (in grams) achieved by the forelimbs. Grip strength of each mouse was averaged from 5 consecutive trials.

Acoustic startle response tests were performed in acoustic apparatuses with mouse enclosures (San Diego Instruments, San Diego, CA) in sound-attenuating cubicles. The amplitude of each startle response was measured using a piezoelectric movement–sensitive platform. Acoustic stimuli and steady background noise (70 dB) were delivered through a loudspeaker. Briefly, mice were put in the mouse enclosures and acclimated for 2 min. Six trials of 40-ms 120 dB white noise bursts were then presented with a variable intertribal interval. Vmax values recorded by Acoustic Startle Program (San Diego Instruments, San Diego, CA) were averaged from six startle trials for each mouse.

## Acknowledgements

This work was supported by NIH grants: NS081282 (to ME and KAM), DA036596 (to KAM), EY012287 (to BMW), R01 NS050596, NS073930 (BX), grant 2015R1A2A2A01003413 from the Ministry of Science, Korea (to PLH), Platform for Drug Discovery, Informatics, and Structural Life Science from the Ministry of Education, Culture, Sports, Science and Technology (MEXT), Japan (to HU) and JSPS KAKENHI Grant Number 25893178 (to KS). We thank Ms. Natalia Martemyanova for mouse breeding and genotyping, Dr. Masayoshi Mishina for providing $Gng7^{-/-}$ mice, Dr. Yuqing Li for sharing $Rgs9-Cre$ line and Dr. Denis Herve for providing Gαolf antibodies.

## Additional information

### Funding

| Funder | Grant reference number | Author |
| --- | --- | --- |
| National Institutes of Health | NS081282 | Michelle E Ehrlich<br>Kirill A Martemyanov |
| National Institutes of Health | EY012287 | Barry M Willardson |
| National Institutes of Health | NS073930 | Baoji Xu |
| Ministry of Science, ICT and Future Planning | 2015R1A2A2A01003413 | Pyung-Lim Han |

The funders had no role in study design, data collection and interpretation, or the decision to submit the work for publication.

### Author contributions

KX, IM, CCS, YC, Acquisition of data, Analysis and interpretation of data, Drafting or revising the article; KS, PLH, HU, CWD, MEE, Drafting or revising the article, Contributed unpublished essential data or reagents; CWJL, Acquisition of data, Contributed unpublished essential data or reagents; BX, Analysis and interpretation of data, Drafting or revising the article; BMW, KAM, Conception and design, Analysis and interpretation of data, Drafting or revising the article

### Ethics

Animal experimentation: This study was performed in strict accordance with the recommendations in the Guide for the Care and Use of Laboratory Animals of the National Institutes of Health. All procedures were approved by the Institutional Animal Care and Use Committee (IACUC) protocol (#14-001) at The Scripps Research Institute.

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
