## [Decision Letter]

Thank you for submitting your work entitled "Stable G protein-effector complexes in striatal neurons: mechanism of assembly and role in neurotransmitter signaling" for peer review at *eLife*. Your submission has been favorably evaluated by a Senior editor, a Reviewing editor, and two reviewers.

The reviewers have discussed the reviews with one another and the Reviewing editor has drafted this decision to help you prepare a revised submission.

Summary:

The concept of a signaling complex in the striatum is a fascinating observation worthy of publication in *eLife*. However, there was concern about some of the data that needs to be addressed prior to publication.

Essential revisions:

1) The co-IP expt in Figure 1 needs a specificity control. One way would be to use AC5^–/–^ mouse. The lysate lanes should be shown for both WT and KO co-IP.

2) The BRET data need to be explained. Why is haloperidol reversing an effect through D1Rs? As it stands, these results are troubling and possibly mis-interpreted.

3) The interpretation of data shown in Figure 3 needs to be clarified. The effect could all be explained by enzyme abundance change based on the previously known chaperone function of PhLP and not requiring complex or effects of G protein-GDP form as concluded.

4) More detail related to these concerns can be found below.

*Reviewer #1:*

Chaperone proteins have been found to play important roles in facilitating the formation of signaling complexes that could include receptors, G protein subunits and effectors. The present data suggest that the PhLP1 chaperone may play a central role in the assembly of AC5-G_olf_ complex and in their signaling. This is a novel finding and has potentially important clinical implications.

The most striking data came from the PhLP1 conditional KO studies. Although the evidence for the role of PhLP1 in the assembly of AC5-G_olf_ complex is indirect in these genetic data, they have convincingly demonstrated that PhLP1 deficiency leads to severely impaired G_olf_-AC5 signaling and motor skill learning but not locomotion.

While the genetic data support the significance of PhLP1 in receptor signaling in vivo, it is often expected that in vitro biochemical studies could provide direct evidence for protein-protein interactions. Here the in vitro biochemical data overall provide further support for the proposed model. However, none of the data provide direct evidence. There are no data showing direct interactions between PhLP1 and AC5 or Golf. There is no direct evidence of impaired AC5-Golf complex in PhLP1 KO either. Changes in protein expression of Golf and AC5 in PhLP1 KO provide strong, but still indirect evidence. Other mechanisms cannot be ruled out.

Non-cAMP pathways are implicated in both D1 and D2 signaling. Do any of the data suggest that? For example, the lack of change in PhLP1 KO in response to D1 drugs. However, it's not clear why a D1 partial agonist SKF38393 was used.

AC1 and Gs are highly expressed in the striatum during embryonic and early postnatal development. Therefore cAMP signaling may not be impaired in the PhLP1 KO during development. Does this fact bring new interpretations to the knockout phenotype (for example, morphological changes in the striatum)?

*Reviewer #2:*

This paper claims that AC5, a highly expressed AC isoform in striatum, is in stable complex with Gβ2γ7 in vivo and this is required for stability of AC5. It also claims that AC5 stability requires stable association with α, β and γ together in complex, that PhLP1 is necessary to form this complex, and that disrupting the complex by PhLP1 knockout reduces striatal cAMP signaling and produces behavioral deficits. This leads the authors to propose a large signalosome containing all of these proteins and also RGS9, with assembly of this larger signalosome dependent on PhLP1.

The authors describe an interesting and quite extensive set of experiments showing that Golf supports AC5 protein expression, likely by increasing protein stability, and that AC5 has a converse effect on Golf abundance. They show that PhLP1 over-expression increases protein abundance in HEK2 cells. PhLP1 knockout in striatum causes the reverse, and also reduction in RGS9-2. Using this mouse they observe major cytological defects in striatum, reduced cAMP production in striatal membrane fractions, and deficits in motor behaviors.

I think that this is an interesting series of experiments that convincingly shows that PhLP1 is important to support Golf, AC5 and RGS9 protein abundance in striatum. On the more critical side, I personally like the larger signalosome idea but am not convinced of it from the data shown. As far as I can tell, all of the effects observed in the PhLP1 KO are consistent with changes in overall protein abundance from co-chaperone actions of PhLP1 in beta-gamma and beta-RGS9 complex formation. The effect on AC5 abundance might indicate that it is included in the complex but might also occur secondarily through changes in Golf subunit abundance. I also think that the signaling data seem consistent with effects of protein abundance changes rather than as evidence that signaling must occur from the larger complex. The knockout mouse is very interesting but there, also, I think that the effects might be explained in a simpler way. The behavioral deficits are interesting but quite complicated to interpret, due to the cytological defects noted.

1) The only direct evidence for the protein complexes claimed is in the co-IP shown in Figure 1. This lacks important controls. The simplest worry is that AC5 IgG could just be sticky compared to control IgG. Different batches of non-immune serum can vary widely in nonspecific binding. One can even get tricked by pre-immune serum from the same animal due to inflammation (acute phase effect). Here the investigators have an AC5 KO mouse. In Figure 1 there should be a variant of Figure 1 co-IP with AC5^–/–^ striatum. Alternative would be to use HEK2 cells to ask, at least in that system, how much α, β and γ come down in the absence of AC5 over-expression. If I understand correctly, one would also expect non-hydrolyzable GTP analogue to partially dissociate the complex but this is not shown or discussed.

2) Figure 2 reports a BRET assay designed to detect Golf activation based on free beta-gamma binding to a NanoLuc-labeled GRK2. 10 μM dopamine caused a BRET change that was reversed by 100 μM haloperidol. However, haloperidol is a D2 and not D1 antagonist, and D1 but not D2 receptors couple to Golf. Of course 100 μM is a huge concentration of haloperidol, but so is 10 μM of dopamine. Based on simple competition, I don't think a 10-fold excess of haloperidol will displace dopamine at D1 receptors. This makes think that the BRET results are spurious and incorrectly interpreted.

3) The increased basal AC activity described in Figure 3 is proposed to represent enzyme activation by free α-GDP subunits due to PhLP1 over-expression. I don't understand this logic because AC5 protein levels are increased. Isn't a simpler explanation that the basal activity is increased due to more AC5 protein as shown in Figure 3? I also don't see how results shown in Figure 3 control for the observed change in forskolin potency because one would not expect a difference with activated G protein instead of forskolin – here there is not synergy but, if anything, competition that might produce an opposite effect.

---

## [Author Response]

Essential revisions:

*1) The co-IP expt in Figure 1 needs a specificity control. One way would be to use AC5^–/–^mouse. The lysate lanes should be shown for both WT and KO co-IP.*

We performed the control experiment showing the results of AC5 immunoprecipitation from AC5 knockout samples. As expected, we see no Golf pull-down from knockout lysates confirming the specificity of the AC5-G_olf_co-immunoprecipitation. These results are reported in the revised Figure 1.

*2) The BRET data need to be explained. Why is haloperidol reversing an effect through D1Rs? As it stands, these results are troubling and possibly mis-interpreted.*

In our system we use antagonist solely to demonstrate reversibility of the signal, which serves to indicate that the effect of dopamine is mediated by the signaling initiated by the D1R. All of our conclusions are derived from analyzing the maximal responses to agonist (dopamine) application. Although classically defined as D2 antagonist, haloperidol has a reasonable affinity for D1R as well, 63 nM (JBC, 1986, 8397). We performed additional experiments, demonstrating that the BRET signal is specifically associated with the activity of the D1R and exogenous Golf and showed the equivalence of haloperidol to a more selective D1R antagonist SCH39166 in the assays. These experiments are presented in revised Figure 2.

*3) The interpretation of data shown in Figure 3 needs to be clarified. The effect could all be explained by enzyme abundance change based on the previously known chaperone function of PhLP and not requiring complex or effects of G protein-GDP form as concluded.*

We are glad that the reviewers brought up this point, which we agree was not sufficiently discussed. Based on accumulated data, we think PhLP1 affects AC5 activity by a dual mechanism. On the one hand, it upregulates AC5 activity and this effect is likely responsible for the corresponding increase in receptor stimulated cAMP production (compare ~1.7 fold expression upregulation in Figure 3 to ~ 1.5 fold of ISO driven cAMP production in Figure 3). However, the changes in basal or FSK-induced cAMP production (~4-fold) are much greater than the change in the AC5 expression level. This suggests the involvement of other mechanisms by which PhLP1 could regulate AC5 activity. Indeed in panels Figure 3 though 3G we provide evidence that exogenously added PhLP1 can also stimulate AC5 by releasing Gαolf from the inhibitory constraints of Gβγ. We clarified these points in the revised version of the manuscript.

Reviewer #1:

*While the genetic data support the significance of PhLP1 in receptor signaling in vivo, it is often expected that in vitro biochemical studies could provide direct evidence for protein-protein interactions. Here the in vitro biochemical data overall provide further support for the proposed model. However, none of the data provide direct evidence. There are no data showing direct interactions between PhLP1 and AC5 or Golf. There is no direct evidence of impaired AC5-G_olf_complex in PhLP1 KO either. Changes in protein expression of Golf and AC5 in PhLP1 KO provide strong, but still indirect evidence. Other mechanisms cannot be ruled out.*

We appreciate reviewer’s understanding that this report is the first to demonstrate the involvement of PhLP1 in controlling assembly of G protein signaling complexes in the striatum. Establishing the role, this study covers a wide territory from behavioral analysis of mouse behavior to alterations in signaling reactions and biochemical studies of protein complexes. When building the model describing PhLP1 involvement, we rely on previous biochemical studies that established and extensively characterized direct interactions of PHLP1 with Gβγ subunits (PNAS, 2015, 2413; JBC, 2009, 16386; Biochemistry, 1998, 15758). We understand that further studies will likely be needed to examine whether in addition to Gβγ binding, PhLP1 also forms direct interactions with AC5 or Gαolf. Thus, our scheme summarizing the conclusions from this work (Figure 7) is tempered and reflects only what we established beyond reasonable doubt (indirect effects through Gβγ).

*Non-cAMP pathways are implicated in both D1 and D2 signaling. Do any of the data suggest that? For example, the lack of change in PhLP1 KO in response to D1 drugs. However, it's not clear why a D1 partial agonist SKF38393 was used.*

The reviewer is absolutely correct; we did not consider that non-cAMP pathways initiated by the D1R may compensate for the down-regulation of G_olf_-AC5 signaling branch. We discuss this possibility in the revised manuscript. The choice of SKF38393 was dictated by its selectivity for D1 receptor and extensive use in hundreds of similar studies, creating a convenient reference for the expected effects or the lack of thereof.

*AC1 and Gs are highly expressed in the striatum during embryonic and early postnatal development. Therefore cAMP signaling may not be impaired in the PhLP1 KO during development. Does this fact bring new interpretations to the knockout phenotype (for example, morphological changes in the striatum)?*

The Cre driver line that we used to knockout PhLP1 in the striatum (*Rgs9-Cre*) turns on the expression rather late in development, ~ P6 (J Neurosci, 2007, 27:14117). This coincides with the induction of Golf expression and maturation of medium spiny neurons. Thus, we think that morphological changes in the striatum that we observe are related to changes in G_olf_-AC5 signaling. However, mechanisms of morphological changes in striatal neurons were not an emphasis of this report, hence we rather focused on signaling changes in adult brains. However, we agree that examining developmental mechanisms might be an important question to be addressed in future studies.

Reviewer #2:

*1) The only direct evidence for the protein complexes claimed is in the co-IP shown in Figure 1. This lacks important controls. The simplest worry is that AC5 IgG could just be sticky compared to control IgG. Different batches of non-immune serum can vary widely in nonspecific binding. One can even get tricked by pre-immune serum from the same animal due to inflammation (acute phase effect). Here the investigators have an AC5 KO mouse. In Figure 1 there should be a variant of Figure 1 co-IP with AC5^–/–^striatum. Alternative would be to use HEK2 cells to ask, at least in that system, how much α, β and γ come down in the absence of AC5 over-expression. If I understand correctly, one would also expect non-hydrolyzable GTP analogue to partially dissociate the complex but this is not shown or discussed.*

As mentioned in response to “Essential Revisions” point #1, we have conducted the control experiment involving AC5 knockout tissues and report it in revised Figure 1.

*2) Figure 2 reports a BRET assay designed to detect Golf activation based on free beta-gamma binding to a NanoLuc-labeled GRK2. 10 μM dopamine caused a BRET change that was reversed by 100 μM haloperidol. However, haloperidol is a D2 and not D1 antagonist, and D1 but not D2 receptors couple to Golf. Of course 100 μM is a huge concentration of haloperidol, but so is 10 μM of dopamine. Based on simple competition, I don't think a 10-fold excess of haloperidol will displace dopamine at D1 receptors. This makes think that the BRET results are spurious and incorrectly interpreted.*

Please refer to our response to “Essential Revisions” point #2 above.

*3) The increased basal AC activity described in Figure 3 is proposed to represent enzyme activation by free α-GDP subunits due to PhLP1 over-expression. I don't understand this logic because AC5 protein levels are increased. Isn't a simpler explanation that the basal activity is increased due to more AC5 protein as shown in Figure 3? I also don't see how results shown in Figure 3 control for the observed change in forskolin potency because one would not expect a difference with activated G protein instead of forskolin – here there is not synergy but, if anything, competition that might produce an opposite effect.*

As we mention in response to point #3 of “Essential Revisions” ~1.5 fold increase in AC5 expression does not fully account for ~4 fold in AC5 activity upregulation. Panels D, E, F, G of Figure 3 establish that in addition to regulation of AC5 expression, PhLP1 also independently affect its activity by releasing Gαs/olf subunits. In respect to interpretation of data shown in Figure 3, the experiment simply shows that PhLP1 is ineffective when activated Gαs is provided in the assay, because exogenous Gαs-GTP would compete with endogenous Gαolf-GDP released by PhLP1. We clarified this point in the revised manuscript.